# Human Filariasis in Africa (2000–2025): Changing epidemiology, uneven diagnostic progress, and persistent neglect

Wanesa Wilczyńska[1]*, Aymar Akilimali[2], Krzysztof Korzeniewski[1]

1 Department of Epidemiology and Tropical Medicine, Military Institute of Medicine—National Research Institute, Warsaw, Poland, 2 Department of Research, Medical Research Circle (MedReC), Gisenyi, Goma, DR Congo

* wwilczynska@wim.mil.pl

## Abstract

### Background

Human filariases remain a major group of neglected tropical diseases (NTDs) in Africa. Despite large-scale control strategies and availability of effective drugs, persistent and re-emerging infections indicate gaps in surveillance and diagnostic capacity. Over the past two decades, diagnostic technologies and elimination programs have evolved, yet no synthesis has assessed how these changes affected research output and prevalence estimates across the continent. The objective of this study was to systematically review and synthesize population-based studies on human filariases in Africa from 2000 to 2025, examining temporal trends, diagnostic methods, and prevalence patterns across species and regions.

### Methods

A systematic review compliant with PRISMA 2020 was conducted for studies published between January 2000 and October 2025 in PubMed, Scopus, ScienceDirect and African Index Medicus. We included population-based studies reporting prevalence of human filariases using confirmatory diagnostics. Data extraction included study characteristics, diagnostic method and prevalence estimate. Study quality was evaluated using the Newcastle–Ottawa Scale.

### Results

A total of 180 studies from 31 African countries were included. Research activity peaked in 2011–2015 and then declined. Microscopy remained the dominant diagnostic method throughout the 25-year period, although serology and molecular tools increased after 2011. A consistent reduction in prevalence was observed for lymphatic filariasis and onchocerciasis in settings with mass drug administration (MDA). Mansonellosis and loiasis showed no comparable decline and were frequently

**Data availability statement:** The analyzed data presented in this study are available in S2 Appendix and in the Zenodo repository, https://doi.org/10.5281/zenodo.17608781.

**Funding:** The APC was funded by the Military Institute of Medicine–National Research Institute, Warsaw, Poland. The funders had no role in study design, data collection and analysis, decision to publish, or preparation of the manuscript.

**Competing interests:** The authors have declared that no competing interests exist.

detected incidentally, reflecting limited diagnostic oversight and the absence of targeted elimination programs.

## Conclusions

Long-term MDA programs are associated with reduced prevalence of lymphatic filariasis and onchocerciasis. Loiasis shows a increasing and unstable trend, influenced by diagnostic method and lack of targeted interventions. Mansonellosis remains neglected, underdiagnosed, and largely unmonitored. Strengthening local diagnostic capacity and integrating filariasis screening into existing platforms (e.g., malaria programs) may prevent silent transmission and resurgence in areas declared free of infection.

### Author summary

Human filariases, including lymphatic filariasis, onchocerciasis, loiasis, and mansonellosis, remain important neglected tropical diseases across Africa. Despite large-scale control programs and the availability of effective drugs, infections persist and sometimes re-emerge, partly due to limited surveillance and diagnostic capacity. In this study, we systematically reviewed 180 population-based studies published between 2000 and 2025, examining prevalence trends, diagnostic methods, and research activity across 31 African countries. Our findings show that long-term mass drug administration programs have reduced lymphatic filariasis and onchocerciasis in treated areas. In contrast, loiasis and mansonellosis. remain underdiagnosed and largely unmonitored, reflecting the lack of targeted elimination strategies. By highlighting gaps in diagnostics and surveillance, this work underscores the need to strengthen local diagnostic capacity and integrate filariasis monitoring into existing platforms, such as malaria programs. These insights can guide public health strategies to prevent silent transmission and resurgence, ultimately supporting more effective elimination of filarial infections in Africa.

## Introduction

Human filariases are vector-borne infections caused by nematodes of the family Onchocercidae. Classified by World Health Organization (WHO) as neglected tropical diseases (NTDs), they disproportionately affect rural and socio-economically marginalized communities where access to diagnostics, treatment, and surveillance is limited [1,2]. Although effective and inexpensive drugs exist, these diseases continue to cause chronic morbidity, restricted mobility, social stigma, and significant economic losses in affected households [2].

Four filarial infections are of major relevance in Africa: lymphatic filariasis (LF), onchocerciasis, loiasis, and mansonellosis [2,3]. LF, primarily caused by *Wuchereria bancrofti*, affects an estimated 70 million people globally, nearly 40% of whom live in

Sub-Saharan Africa [3]. Onchocerciasis (*Onchocerca volvulus*) remains one of the leading infectious causes of preventable blindness; globally, approximately 25 million people are infected, of whom an estimated 300,000 have lost their vision and nearly one million suffer from visual impairment [1,3]. Loiasis (*Loa loa*), endemic in Central and West Africa, affects more than 10 million people on the continent and poses unique challenges for elimination programs High microfilaremia is associated with severe and sometimes fatal adverse events following ivermectin administration, creating "treatment exclusion zones" that limit mass drug administration (MDA) [3,4]. In contrast mansonellosis caused mainly by *Mansonella perstans* and *M. streptocerca*, rarely *M. ozzardi* — is estimated to affect over 114 million people in Africa, making it the most widespread filarial infection on the continent [3]. Although previously assumed to be clinically mild, emerging data suggest associations with chronic inflammatory symptoms and interference with serological tests, further complicating differential diagnosis in endemic areas [5,6].

In several regions, these pathogens occur simultaneously, sharing vectors, ecological niches, and risk factors [7]. Co-endemicity — especially involving loiasis creates operational and ethical barriers for elimination campaigns [8]. At the same time, mansonellosis often remain invisible to national surveillance systems, reinforcing epidemiological blind spots [6]. As a result, official prevalence estimates may significantly underestimate the true burden of disease.

Historically, diagnosis in endemic countries has relied on microscopy, due to low cost and minimal infrastructure requirements. However, microscopy has limited sensitivity in low-intensity infections and cannot reliably differentiate between species such as *M. perstans* and *L. loa*. Over the past decade, serological methods (antigen detection tests for LF and onchocerciasis) and molecular techniques (PCR, LAMP) have increasingly been used in research settings, improving detection accuracy and revealing infections previously missed by routine surveillance [9,10]. These diagnostic shifts fundamentally influence prevalence estimates — declines in microscopy-based studies may not necessarily reflect real reductions in transmission [11].

Despite nearly 25 years of control programs — including the Global Programme to Eliminate Lymphatic Filariasis (GPELF) and the African Programme for Onchocerciasis Control (APOC) — major gaps persist. Surveillance remains fragmented, particularly in areas without laboratory infrastructure [12–14]. Many countries lack capacity for molecular diagnostics, and research output is unevenly distributed, heavily concentrated in a small number of academic institutions [11,15].

Although numerous studies have examined specific pathogens or regions, no synthesis has evaluated how diagnostic practices and research intensity have evolved across Africa and how these changes influence reported prevalence. This creates uncertainty about whether observed epidemiological patterns reflect true reductions in disease burden or methodological artefacts. This systematic review addresses that gap by examining temporal, geographical, and methodological trends in studies reporting prevalence of lymphatic filariasis, onchocerciasis, loiasis and mansonellosis in Africa between 2000 and 2025. By linking prevalence estimates with diagnostic methods and research activity over time, we provide a continent-wide perspective on progress, persistent inequities, and blind spots that continue to hinder filariasis elimination.. Therefore, the objective of this systematic review was to evaluate how diagnostic practices and research intensity have evolved across Africa and how these changes influence reported prevalence of lymphatic filariasis, onchocerciasis, loiasis and mansonellosis.

## Methods

### Search strategy

This systematic review followed the PRISMA 2020 guidelines [16] and was designed to identify, synthesize, and analyse epidemiological and methodological trends in human filariasis research across Africa from 2000 to 2025, focusing on lymphatic filariasis, onchocerciasis, loiasis, and mansonellosis.

We searched PubMed, Scopus, ScienceDirect, and African Index Medicus (AIM) for articles published between January 1, 2000, and October 8, 2025, using combinations of keywords and MeSH terms related to filariasis, prevalence, Africa, and individual filarial species. Reference lists of included articles and relevant reviews were manually screened to identify additional studies. The complete search strings and database outputs are provided in S1 Appendix.

This systematic review was conducted according to a pre-defined protocol; however, the protocol was not registered in PROSPERO.

## Eligibility criteria

Studies were selected using the PECO framework [17]:
Population: Human populations in African countries;
Exposure: Exposure to filarial parasites or vectors;
Comparator: Trends over time and across diagnostic methods and regions; comparison is implicit rather than experimental;
Outcomes: Prevalence estimates of filarial infections.

We included population-based, cross-sectional studies published in peer-reviewed journals in English, with full-text access, and reporting both sample size and infection counts. Excluded were: animal studies, case reports, reviews, editorials, programmatic reports, preprints, and studies without sufficient data. Additionally, studies relying solely on self-reported symptoms or questionnaire-based assessments (e.g., history of Calabar swellings or eye worm passage) were excluded, as these methods lack confirmatory diagnostic evidence. Studies focusing exclusively on entomological surveillance, animal reservoirs, or modeling without empirical prevalence data were excluded.

Eligible studies were population-based investigations reporting human filarial prevalence with confirmatory diagnostics. This included surveys conducted in diverse settings, such as schools, households, and communities, as well as cluster-based surveys evaluating the impact of mass drug administration programs, provided that original prevalence data were reported with sufficient methodological detail. Studies relying solely on self-reported symptoms were excluded. Studies focusing exclusively on entomological surveillance, animal reservoirs, or modeling without empirical prevalence data were excluded. Entomological studies were intentionally excluded from the systematic synthesis, as the review focused on human prevalence estimates. However, key entomological evidence is discussed narratively to provide biological and ecological context for the observed epidemiological patterns.

## Study selection and Data Extraction

Search results were imported into Rayyan software [18] to remove duplicates and screen records. Two reviewers independently screened titles, abstracts, and full texts; disagreements were resolved by consensus or third-party adjudication. A standardized data extraction form (Microsoft Excel) was used to collect: first author, publication year, data collection period, country, setting, sample size, diagnostic method, pathogen, prevalence, and participant age group (Table A in S2 Appendix). For the purpose of this review, a data point was defined as a unique combination of pathogen, study, population, and diagnostic method. When studies reported data for multiple filarial species, each species was treated as an independent data point, ensuring that prevalence estimates were not duplicated. If a study reported multiple datapoints from the same site or timepoint (e.g., different age groups or diagnostic methods), these were extracted separately to capture methodological variation, and overlapping prevalence estimates for the same population and period were aggregated using the arithmetic mean for the analysis of temporal trends.

For the analysis of temporal trends, the year of sample collection was used. If the data collection spanned several years, the arithmetic mean of the collection years was calculated. If the midpoint of data collection overlapped two five-year intervals, the study was assigned to the interval covering the majority of its data collection period. In the absence of information on the date of data collection, the year of publication was used as a proxy.

## Quality assessment

Study quality and risk of bias were assessed using the Newcastle-Ottawa Scale (Table A in S3 Appendix) [19,20], evaluating:

• Selection of study population (up to 4 stars),

- Comparability of study groups (up to 2 stars),

- Outcome assessment (up to 3 stars).

Studies scoring ≥6 stars were considered of moderate to high quality. Each study was rated independently by two reviewers, with discrepancies resolved through discussion.

## Data synthesis and statistical analysis

Descriptive statistics were used to summarize study characteristics including sample size, diagnostic method, and species distribution. Mean prevalence values were calculated for five-year intervals to visualize temporal trends; these means are unweighted by sample size to provide a simple overview of reported prevalence across studies. Random-effects meta-analyses were additionally performer using the metaprop function in the meta package in R [21,22] to estimate pooled. Due to substantial heterogeneity in study designs, diagnostic methods, population characteristics, and the extended 25-year study period, the pooled prevalence estimates were not included in the main text, as their inclusion could be misleading. Instead, the full quantitative results, including forest plots, subgroup analyses, and summary tables, are provided in supplementary materials (Tables A-C and Figs A-D in S4 Appendix) as a statistical reference. Studies with incomplete or missing data were excluded from the analysis. All extracted variables (study characteristics, diagnostic method, prevalence, participant age group, etc.) were compiled in a standardized Excel form prior to analysis. Maps and pie charts were generated in R (packages ggplot2, sf) and Microsoft Excel to illustrate geographic distribution and relative species prevalence.

## Results

### Study selection

A total of 10,122 records were retrieved from four databases. After removing 4,368 duplicates, 5,754 records were screened by title and abstract. Of these, 296 full-text articles were reviewed for eligibility, and 180 met the inclusion criteria and were included in the final synthesis (Fig 1).

### Characteristics of included studies

The 180 included studies were published between 2000 and 2025 and covered data from 31 African countries. Most of them focused on local populations (both children and adults; 64.3%), while 15.5% investigated only children and another 14.6% only adults. The remaining studies did not provide detailed information on the age of participants. Across all studies, the total number of participants was 745,568. From the 180 studies, a total of 328 distinct data points were extracted.

LF represented the largest proportion of studies (44.5%), followed by onchocerciasis (34.8%), loiasis (12.2%), and mansonellosis (8.5%). Multi-species studies investigating more than one filarial pathogen accounted for 22.2% of the dataset.

The most frequently represented countries were Cameroon, Nigeria, Gabon, and Equatorial Guinea. In contrast, seven countries had only one or two published studies, and 23 had none. The geographic distribution of studies was highly uneven, with a marked research concentration in West and Central Africa (Fig 2).

### Temporal trends and methodological evolution

Between 2000 and 2009, only 1 study were published annually on average, increasing to 8.5 per year between 2010 and 2019 and 12 per year between 2020 and 2025. This rise reflects both expanded surveillance efforts and greater use of molecular diagnostic techniques in recent years.

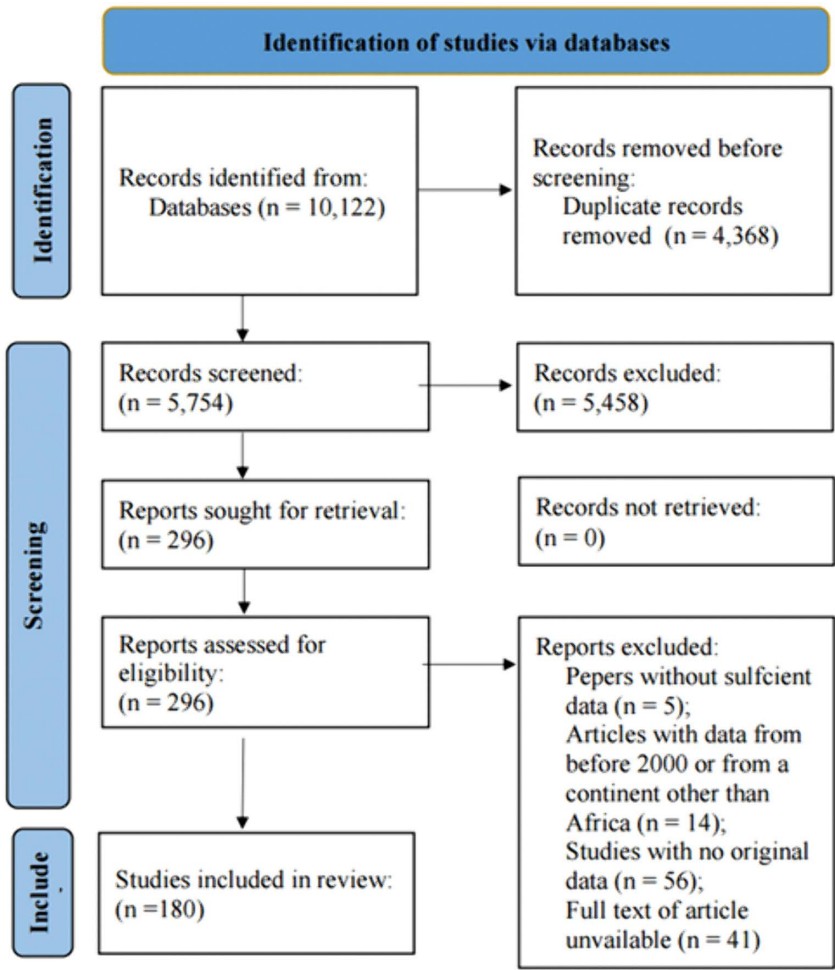

**Fig 1. PRISMA flow diagram illustrating the process of identification, screening, eligibility assessment, and inclusion of studies on human filarial infections (2000–2025).**

Microscopy was the dominant diagnostic method in early years (2000–2010), accounting for 55% of studies. However, from 2018 onward, the use of serological (e.g., ICT, FTS, Ov16 ELISA) and molecular assays (PCR, qPCR) increased, comprising nearly 60% of studies in the most recent period. Temporal trends were examined across all diagnostic modalities. While microscopy and molecular methods provide evidence of active infection, serological assays reflect exposure. We formally tested the effect of diagnostic method on prevalence estimates using meta-regression limited to studies of active infections (microscopy vs. molecular), and no significant differences were observed across species (Table B in S4 Appendix Table 2). Serological results were not included in this comparison because they reflect exposure rather than active infection. Combining these data points allows a comprehensive, long-term view of research activity and observed prevalence patterns, acknowledging that short-term studies often capture only limited snapshots of infection dynamics. Pie charts overlaid on prevalence points indicate the proportion of studies using each diagnostic method in each period, providing context for interpreting trends and methodological evolution.

The following sections present species-specific analyses of original studies that met the inclusion criteria. Each subsection summarizes temporal trends, diagnostic approaches, and prevalence patterns based on eligible data points.

## Number of Filariasis Studies in Africa (2000–2025)

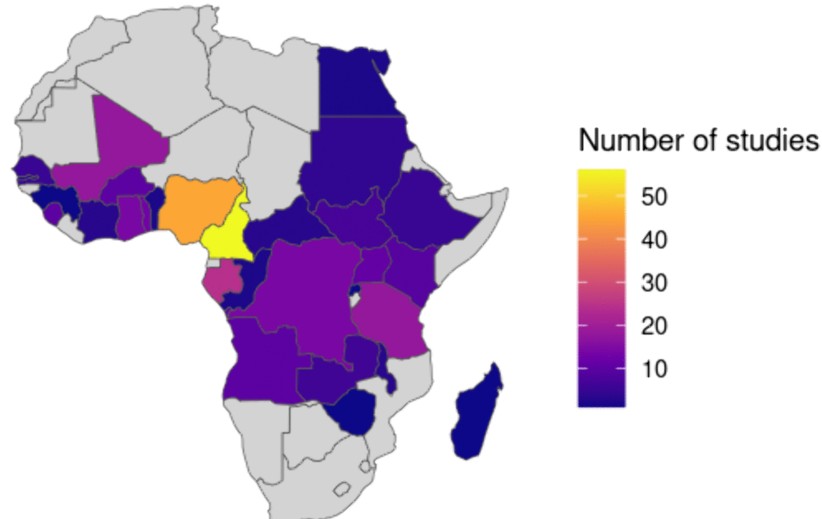

**Fig 2. Geographical distribution of original studies on human filarial infections in Africa (2000–2025) that met the inclusion criteria of this review.** Base map layer: Natural Earth (Admin 0 – Countries, 1:10m). Source: https://www.naturalearthdata.com/downloads/10m-cultural-vectors/10m-admin-0-countries/. Terms of use: https://www.naturalearthdata.com/about/terms-of-use/b(public domain).

Although random-effects pooled prevalence estimates were computed, the main Results focus on descriptive synthesis, while full meta-analytic outputs (pooled estimates, 95% CI, I²/τ², forest plots) are reported in the Table 2 in S4 Appendix (Figs 1–4) due to high heterogeneity across studies

### Lymphatic filariasis (*Wuchereria bancrofti*)

Original studies on LF were conducted in multiple African countries, with the highest research activity recorded between 2011 and 2015. The number of eligible publications declined markedly thereafter. Diagnostic approaches were diverse, with serological tests detecting circulating filarial antigen being the most common, followed by microscopy of blood smears and, less frequently, molecular techniques (conventional PCR, qPCR, nested PCR). Average prevalence values decreased steadily from 2000 to 2025—from approximately 10% in 2000–2005 to below 1% in 2021–2025. Studies were concentrated mainly in Central and Eastern Africa, with the largest number of reports from Nigeria and Tanzania (Figs 3 and 7a).

### Onchocerciasis (*Onchocerca volvulus*)

Studies on onchocerciasis meeting the inclusion criteria were identified in 21 African countries, with the highest number of reports from Nigeria and Cameroon. The number of eligible original studies increased steadily from 2000 onwards, reaching its maximum between 2016 and 2020, after which a decline was observed. Early research (2000–2010) relied predominantly on microscopy of skin snips, while after 2011 additional methods such as serological assays (e.g., Ov16 IgG4) and molecular tests (qPCR, F-RT-PCR) were incorporated. Throughout the study period, microscopy remained the most frequently used approach. Reported prevalence rates declined after 2005 and remained relatively stable for approximately 15 years at an average level of about 5%, followed by an increase observed after 2020 (Figs 4 and 7b).

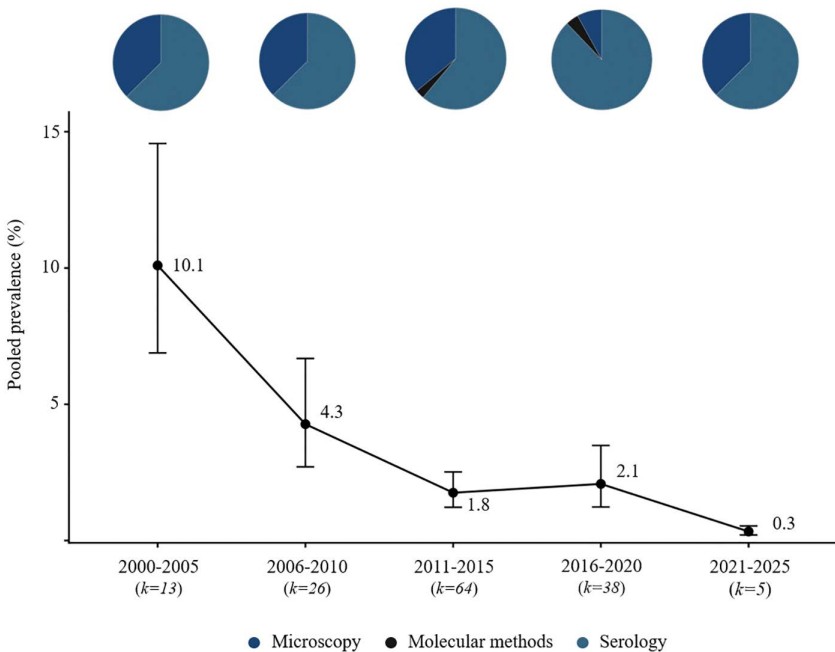

**Fig 3. Temporal trends and diagnostic methods, lymphatic filariasis in Africa (2000–2025).**

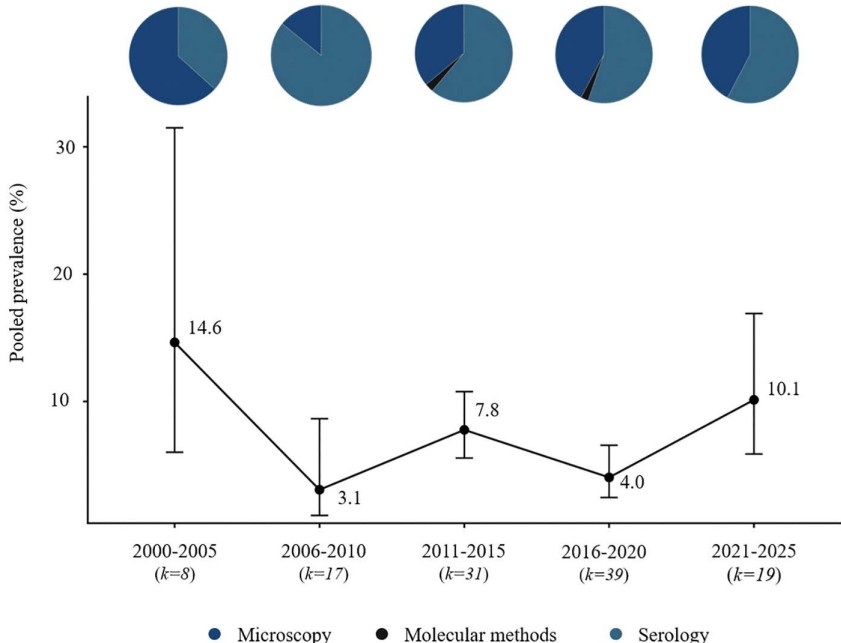

**Fig 4. Temporal trends and diagnostic methods onchocerciasis in Africa (2000–2025).**

## Loiasis (*Loa loa*)

Among the 180 original studies included in this review, research focusing on loiasis was reported from seven African countries, with most studies originating from Cameroon, Gabon, and Nigeria. The number of eligible publications increased until 2015, peaking between 2011 and 2015, followed by a gradual decline in subsequent years. Diagnostic approaches evolved over time: during 2000–2010, microscopy of blood smears was the only method used, while between 2011 and 2020 molecular assays such as qPCR, F-RT-PCR, and nested PCR began to appear. After 2021, serological methods (Pan-IgG-SXP-1) were also introduced. Across the 25-year period, microscopy remained the predominant technique. The highest average prevalence values were recorded in 2006–2010, followed by a marked decline during the next decade and a moderate rise in the most recent years (Figs 5 and 7c).

## Mansonellosis (*Mansonella* spp.)

Research on mansonellosis infections accounted for a smaller proportion of the included studies but followed a similar temporal pattern to that observed for *W. bancrofti*. The number of eligible publications increased between 2006 and 2015, peaking during 2011–2015, and declined thereafter. The majority of studies relied on microscopy for parasite detection, with molecular techniques (qPCR, F-RT-PCR) reported sporadically between 2006 and 2020. The highest prevalence values were observed during 2011–2015, coinciding with the period of greatest research output. Studies originated mainly from Central African countries, particularly Cameroon and Gabon (Figs 6 and 7d).

Where species-level identification was reported, *M. perstans* was the most commonly detected species. However, a substantial proportion of studies reported only *Mansonella* spp. without species-level resolution, reflecting limitations in routine diagnostics and reporting

Overall, the analysis revealed distinct temporal and methodological patterns across filarial species, with notable shifts in diagnostic approaches and variations in the geographic scope of research.

Across the surveyed countries, observed prevalence of lymphatic filariasis, onchocerciasis, and loiasis largely reflected the ESPEN-classified endemicity status. Where MDA had been implemented, prevalence was generally lower than in

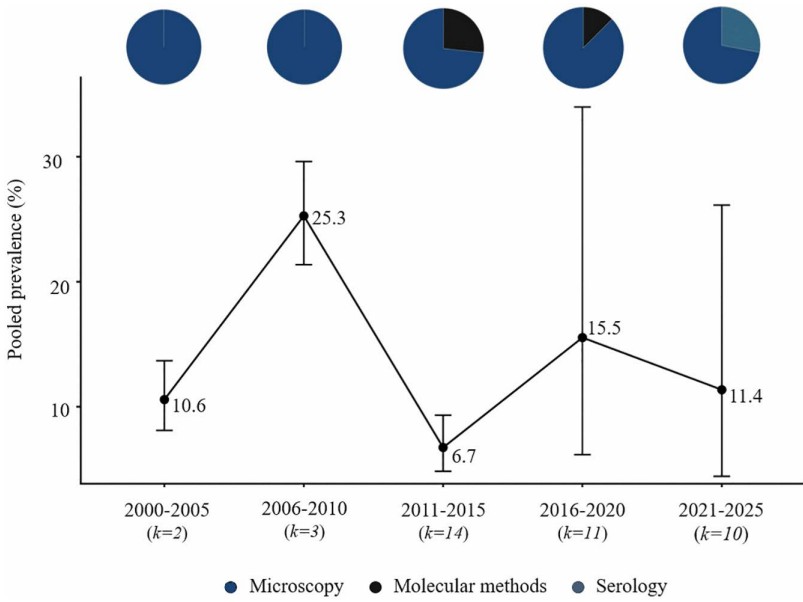

**Fig 5. Temporal trends and diagnostic methods loiasis in Africa (2000–2025).**

**PLOS Neglected Tropical Diseases**

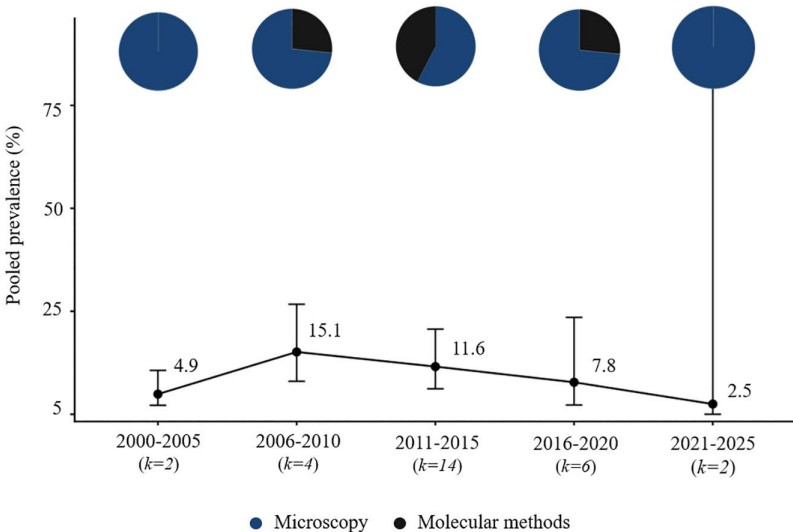

**Fig 6. Temporal trends and diagnostic methods mansonellosis in Africa (2000–2025).**

areas without MDA, although variation existed between regions and time periods. Notably, some areas with reported full MDA coverage still showed substantial prevalence, suggesting incomplete drug uptake or residual transmission foci. Detailed prevalence estimates for each country and parasite are provided in Table C in S4 Appendix

## Disscussion

Our analysis revealed a marked geographic imbalance in where filariasis research has been conducted in Africa over the past 25 years. Although ecological conditions favorable to transmission — high humidity, the presence of competent vectors (mainly *Anopheles*, *Culex*, and *Simulium*) and large rural areas with limited sanitation infrastructure — occur across much of the continent [23–25], most published research originates from West, Central, and, to a lesser extent, East Africa. These regions are characterized by persistent vector presence and intense, ongoing transmission of parasitic diseases. Entomological data and vector habitat mapping published by WHO shows that nearly the entire African continent lies within the potential transmission zone for filarial infections, even if active epidemiological surveillance is not conducted everywhere [26]. Northern Africa and large parts of Southern Africa were nearly absent from research outputs. Several factors overlap here. First, public health priorities in these regions are different — noncommunicable diseases and infections with distinct epidemiological profiles dominate [27], reducing the pressure to invest in filariasis diagnostics and surveillance. Second, climatic conditions in the north (semi-arid and Mediterranean zones) are less favorable for stable vector populations, and thus for sustained transmission [26]. Third, profound disparities in research and laboratory infrastructure persist. For decades, West and Central Africa have hosted large international control and elimination programs (APOC, GPELF, later ESPEN), which not only implemented interventions but also generated research datasets. Funded projects often translate into scientific publications [28]. Where programs were absent, data are absent.

Unequal access to research funding and diagnostic laboratories is a major driver of geographic bias — repeatedly highlighted in bibliometric studies on scientific production in the Global South [29–31]. Countries with stronger research capacity — such as Cameroon, Nigeria or Gabon — produce research data far more frequently than countries where laboratory infrastructure is limited to basic clinical diagnostics [32]. In many settings, field data are never published in indexed journals (PubMed, Scopus) or appear only in local journals, frequently in French or Arabic, rendering them invisible to standard literature searches. This represents a form of publication bias — not against negative

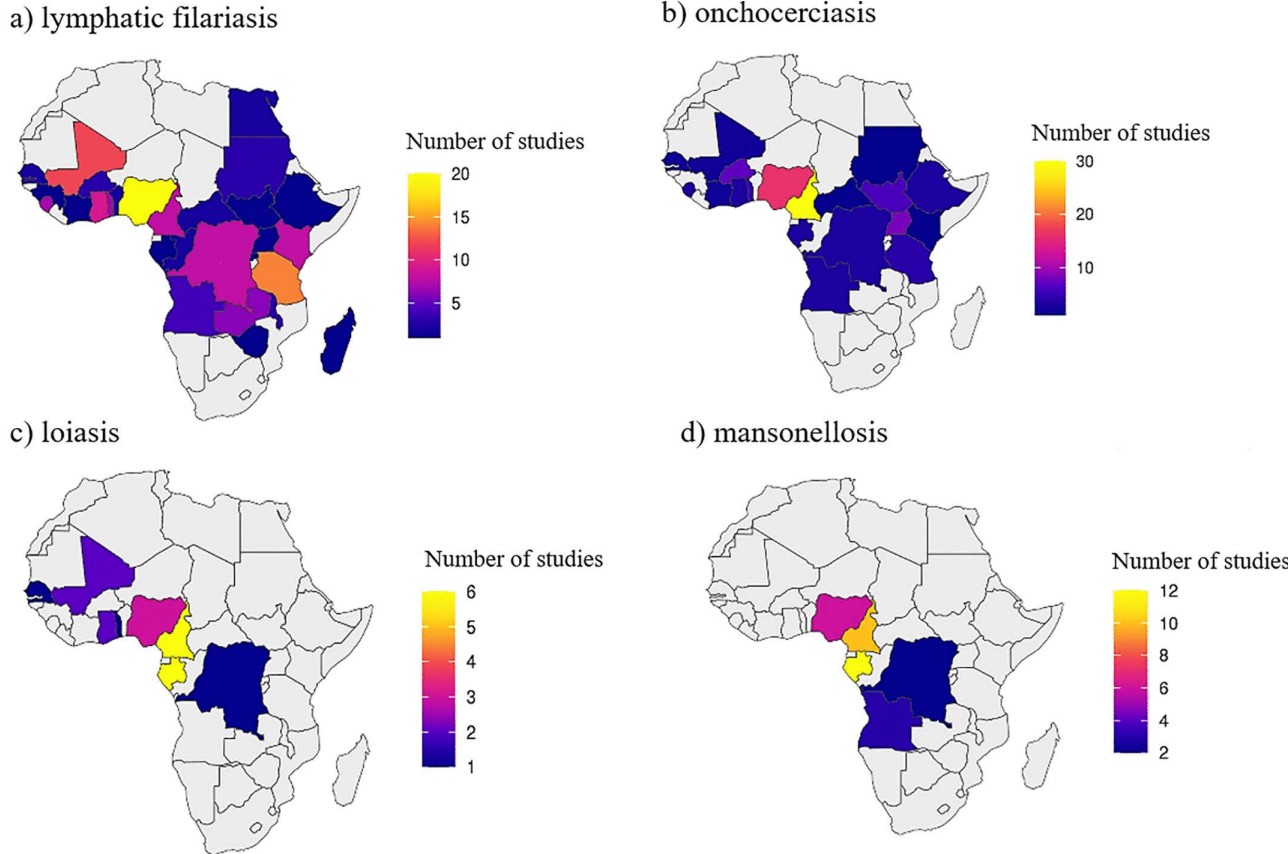

**Fig 7. Geographic distribution of studies on (a) lymphatic filariasis, (b) onchocerciasis, (c) loiasis, (d) mansonellosis in Africa (2000–2025).**
Base map layer: Natural Earth (Admin 0 – Countries, 1:10m). Source: https://www.naturalearthdata.com/downloads/10m-cultural-vectors/10m-admin-0-countries/. Terms of use: https://www.naturalearthdata.com/about/terms-of-use/b(public domain).

results, but against geography [33,34]. Importantly, the absence of data does not mean the absence of disease. It means the absence of surveillance. For the communities living in these areas, this translates into silent, undetected transmission without access to treatment. The NTD literature consistently shows that neglected diseases occur where scientific systems are neglected [35]. This underscores the need for targeted capacity building and investment in local surveillance networks to capture the true epidemiological picture.This geographic inequality is also reflected in diagnostic approaches. Across studies, we observed a clear evolution in the diagnostic methods used. According to global recommendations, microscopy remains the primary diagnostic tool for filariasis (night blood smears for *W. bancrofti*, skin snips for *O. volvulus*, and peripheral blood films for *L. loa* and *Mansonella* spp.). Over the past decade, antigen tests — including ICT and FTS for lymphatic filariasis and Ov16 IgG4 serology for onchocerciasis — have become part of global diagnostic standards, both for clinical diagnosis and monitoring elimination programs [36]. In contrast to microscopy and PCR, which confirm active infection, serological assays reflect immunological exposure; antibodies may persist after infection has resolved [37]. Molecular methods (qPCR, nested PCR, F-RT-PCR) are confirmatory tools used mainly in research or high-sensitivity screening. In routine clinical practice in many African countries, advanced diagnostics remain inaccessible. Laboratories often lack essential equipment, stable electricity, cold chain for reagents, and systems for sample storage. Under such conditions, microscopy becomes the first-line tool — not

because it is optimal, but because it is available [38–40]. Our data reflect this reality: between 2000 and 2010, microscopy dominated in more than half of all studies. Serology and PCR began appearing only after 2011, coinciding with increased availability of rapid tests and elimination programs supported by international partners. Despite this, microscopy remained dominant across nearly all filarial species throughout the 25-year period. It is important to recognize that microscopy has diagnostic limitations — morphological similarities among microfilariae and operator-dependent interpretation may lead to misidentification, which in some regions has direct therapeutic implications (e.g., risk of severe adverse reactions to ivermectin in individuals with high *L. loa* microfilaremia) [36,41]. Meta-analytic comparisons between microscopy and molecular diagnostics show no significant differences in prevalence estimates, confirming that historical microscopy data remain informative for assessing long-term trends. Therefore, in resource-limited settings, microscopy can determine patient care. The combination of epidemiological history, clinical assessment, and basic microscopy enables detection of active infections and decisions regarding treatment referral. In contexts where molecular and serological methods are unavailable, microscopy is not a "lesser method" — it is the backbone that keeps fragile healthcare systems capable of responding to patient needs.

Although entomological studies were not included in the systematic synthesis, available vector surveillance data support the biological plausibility of the observed human prevalence patterns. Entomological research conducted across West, East and Central Africa consistently demonstrates the persistence of competent vectors for filarial transmission, including Anopheles and *Culex* mosquitoes for lymphatic filariasis [42–44] *Simulium* spp. for onchocerciasis [45,46], and *Chrysops* spp. for loiasis [47]. In several endemic regions, vector presence and infection rates remain detectable even in areas where human prevalence has substantially declined following long-term mass drug administration [45,46]. This apparent discordance has been reported particularly for onchocerciasis and lymphatic filariasis and does not necessarily indicate failure of interventions, but rather reflect the different temporal dynamics captured by human and entomological indicators. While human prevalence responds relatively rapidly to repeated MDA, vector populations and their infection rates may persist longer, especially in ecologically favorable settings [48]. Importantly, entomological indicators are rarely integrated into routine surveillance outside well-funded elimination programs, and their availability is even more uneven than that of human prevalence data. As a result, entomological evidence remains fragmented and difficult to synthesize at a continental scale. For this reason, and to preserve methodological coherence, vector-based studies were not included in the systematic analysis, but they remain essential for local risk assessment and for interpreting post-elimination surveillance data.

The adoption of improved diagnostics enabled more accurate monitoring of elimination interventions [38,40]. In the early 2000s, LF was highly endemic across sub-Saharan Africa — in some areas, prevalence exceeded 20–30%, particularly in West and East Africa [49]. Our review demonstrates a clear decline: mean prevalence decreased from >10% in 2000–2005 to <1% after 2020. This decline coincides with the implementation of the GPELF, launched by WHO in 2000— one of the largest public health interventions in the history of neglected tropical diseases. A key strategy was MDA, consisting of annual distribution of albendazole with ivermectin (or diethylcarbamazine outside onchocerciasis-endemic areas). In 2017, WHO introduced the IDA regimen (ivermectin + DEC + albendazole), further improving elimination effectiveness. Monitoring relied on antigen detection (ICT/FTS), allowing assessment of active or recent infection and guiding decisions to stop MDA [50–52]. The program's impact is tangible: Togo was the first African country to declare LF elimination (2017) [53]; Malawi stopped MDA and entered surveillance [53]; Tanzania achieved >90% reduction in infections [54]; in Nigeria — the country with the largest population at risk — transmission has been interrupted in multiple regions [55]. Our data mirror WHO findings, showing substantial reductions in LF prevalence in many African countries where MDA has been implemented across successive rounds, despite substantial logistical barriers, lack of laboratory infrastructure, shortages of personnel, unreliable power supply, and the need to reach remote communities cut off during rainy seasons. These patterns highlight the importance of operational logistics and collaboration with local communities alongside the use of available diagnostics.

Progress in onchocerciasis elimination has been less linear. While transmission has been substantially reduced, hotspots persist (e.g., Cameroon) [56]. The history of onchocerciasis control illustrates how success depends on logistics and implementation strategies. The Onchocerciasis Control Programme (OCP, 1974–2002) focused on vector control through larviciding, eliminating transmission across many West African countries. The subsequent APOC (1995–2015) adopted community-directed ivermectin treatment (CDTI), enabling treatment in areas without health infrastructure [14,57]. In our review, mean prevalence declined from ~15% at the start of the century to ~5% in 2005–2020. After 2020, prevalence appears to rise again (>10%), likely reflecting diagnostic shifts — increased use of Ov16 IgG4 serology, which detects exposure, not necessarily active infection. Progress appears greatest in settings applying dual strategies — vector control + MDA — such as Uganda and Tanzania [14,54]. Implementation remains difficult where conflict, weak infrastructure, or co-endemic loiasis (risk of severe adverse reactions to ivermectin) limit interventions [41,58].

*Loiasis* infections, endemic to rainforest regions of West and Central Africa, illustrate the clinical importance of choosing the correct diagnostic method. Peripheral blood microscopy in the appropriate time window (due to diurnal periodicity) remains the reference standard, as it provides microfilarial density — critical before diethylcarbamazine (DEC) treatment [36,59]. PCR offers superior sensitivity at low microfilaremia but cannot quantify parasite load, limiting standalone use where assessing treatment risk is essential [60]. Serology has extensive cross-reactivity (with *Mansonella* spp., *O. vulvulus*), reducing specificity in endemic settings [60,61]. After the introduction of serology in 2021, our review did not identify a clear increase in pooled laiosis prevalence; instead, prevalence estimates remained consistently high over time. Nevertheless, shifts in diagnostic approaches may influence observed prevalence patterns and should be interpreted cautiously. Diagnostic errors have real clinical consequences — during MDA campaigns, severe neurologic complications occurred in individuals with unrecognized high microfilaremia treated with ivermectin [41,59].

Mansonellosis (*M. perstans, M. streptocerca*, rarely *M. ozzardi*) remain among the most neglected filarial diseases. Although widespread in Africa and parts of South America, they are not classified by WHO as causing significant disability. Without elimination program status, there are no coordinated interventions, no MDA, and epidemiological data remain fragmentary [10]. Diagnosis relies on microscopy, but microfilariae are easily misidentified, especially in mixed infections. Molecular diagnostics clearly differentiate species but are largely restricted to research settings [62]. Clinical manifestations — pruritus, chronic fatigue, musculoskeletal pain, eosinophilia — are nonspecific. For patients, this is an "invisible disease": it rarely causes dramatic complications but impairs quality of life and results in repeated healthcare visits without diagnostic clarity [10,62]. Treatment options are limited, as ivermectin and albendazole show inconsistent efficacy [10]. Mansonellosis is common but invisible — absent from WHO reports yet affecting daily functioning of affected individuals.

Comparison of our findings with ESPEN data on endemicity status and MDA implementation shows general agreement regarding disease distribution and control levels (Table C in S4 Appendix). However, in some cases, prevalence remained relatively high despite reported full MDA coverage. This may reflect discrepancies between reported geographic coverage and actual drug uptake by the population, as well as the influence of historical infection burden and local transmission. These observations highlight the complexity of elimination processes and the importance of incorporating field survey data when interpreting ESPEN reports [15].Despite elimination progress, the true burden of filarial infections remains poorly documented. Access to diagnostics is limited by distance, transport costs, lack of laboratory facilities, political instability, and stigma. As a result, prevalence is systematically underestimated. The cruel paradox is that the populations most affected by parasitic diseases are the least represented in scientific research. Sustained data collection requires stable funding, infrastructure, and trained personnel — resources often absent in endemic regions [32–40]. Moreover, elimination success without continued surveillance can lead to resurgence. Several countries have documented transmission re-emergence after stopping MDA campaigns [63–65]. Research studies conducted by local institutions are essential — unlike programmatic reports, they are not shaped by administrative targets and may detect early warning signals. Future control efforts must reflect the realities of resource-limited settings: fragile infrastructure, workforce shortages, and chronic underfunding [29,30,32,35]. Implementation of advanced, costly methods across all sites is unrealistic. Instead, central

reference laboratories should perform molecular testing on samples collected from peripheral sites. The greatest impact comes from simple, point-of-care antigen and serology tests that require no electricity, no cold chain, and minimal training [38–40]. Integration with existing diagnostics networks for malaria, HIV, and tuberculosis is critical. When blood is already collected — for malaria smears, HIV testing, or on FTA cards during population screening — the same sample can be used for filariasis testing, pending ethical approval. This approach increases the likelihood of detecting silent transmission, especially among asymptomatic carriers who sustain transmission but do not seek care [66,67]. The priority is not technology itself, but ensuring that infections are detected where no one is currently looking — in small clinics, underserved communities, and among children. Strengthening diagnostics requires training, simplified protocols, and improving sample transport — not expensive equipment. Real progress does not require new laboratories; it requires making the most of every resource already available.

An important unresolved question is whether the apparent decline in human prevalence is paralleled by a comparable reduction in vector infection rates. Addressing this would require temporally and geographically matched human and entomological data, which are currently scarce and unevenly distributed across Africa.

### Limitations of the review proces

This review is limited by the availability and accessibility of published data. Many regions of Africa, are underrepresented in indexed databases due to limited research infrastructure, scarce funding, and reduced opportunities for local investigators to publish in high-impact journals. Language barriers and the predominance of non-indexed local journals further restrict data capture [68,69]. Consequently, our synthesis may not fully reflect the true epidemiological situation across the entire continent, and the observed geographic patterns likely reflect both real differences in research activity and structural inequalities in data availability.

Furthermore, this review focuses exclusively on human prevalence data and does not integrate entomological indicators of transmission intensity. While vector infection rates would provide an important complementary perspective, entomological data are highly uneven across regions, methodologically heterogeneous, and rarely collected in parallel with human surveys. Integrating such data into a continent-wide synthesis would risk introducing additional bias rather than resolving uncertainty. Therefore, the analysis is limited to examining how diagnostic practices and research intensity shape the epidemiological picture derived from human data.

Finally, it should be noted that our synthesis includes only studies meeting pre-specified inclusion criteria, i.e., reporting raw prevalence data and using confirmatory diagnostics. Many other publications were excluded for lacking these elements. As a result, while our analysis provides a consistent, long-term view of research trends and prevalence patterns, some geographic regions and time periods may be underrepresented, and the findings should be interpreted in this context

These limitations do not merely reflect gaps in the literature but indicate regions and populations at risk of silent transmission. Strategic prioritization of surveillance in underrepresented areas, integration of simple diagnostics into existing health infrastructure, and fostering local research capacity are essential to close these gaps.

### Conclusions

The literature review highlights a striking contrast between filarial diseases incorporated into global public health programs and those classified as "non-priority." Lymphatic filariasis and onchocerciasis benefit from long-term elimination strategies, which have led to gradual and sustained reductions in prevalence. In contrast, *Mansonella* infections — despite their wide geographic distribution — remain largely invisible to health systems. Epidemiological data are fragmented, diagnostics rely predominantly on microscopy, and effective molecular tools are only rarely implemented. As a consequence, the global picture of disease burden is distorted, and transmission may remain undetected for years.

Across studies, one pattern consistently emerges: the greatest progress occurs in settings where interventions leverage existing health system structures and where long-term engagement of local teams is prioritized. Sustainable progress in filariasis control relies not only on high-tech diagnostics but also on leveraging available resources, training personnel, integrating filarial screening into routine care, and strategically using molecular and serological tools to guide elimination efforts. Flexible use of available resources and integration of diagnostics into routine clinical care appear to be more realistic and sustainable than reliance on costly, technologically demanding solutions.

## Supporting information

**S1 Appendix. Search strategy.**
(DOCX)

**S2 Appendix. Summary of study-level characteristics and reported prevalence from 180 filariasis studies conducted in Africa between 2000 and 2025.**
(DOCX)

**S3 Appendix. Quality assessment and risk of bias evaluation using the Newcastle-Ottawa Scale.**
(DOCX)

**S4 Appendix. Additional statistical calculations and forest plots.**
(DOCX)

**S1 PRISMA Checklist. PRISMA 2020 checklist.** From: Page MJ, McKenzie JE, Bossuyt PM, Boutron I, Hoffmann TC, Mulrow CD, et al. The PRISMA 2020 statement: an updated guideline for reporting systematic reviews. MetaArXiv. 2020, September 14. https://doi.org/10.31222/osf.io/v7gm2. For more information, visit: www.prisma-statement.org
(DOCX)

## Acknowledgments

We are sincerely grateful to Professor Małgorzata Paul for her generous guidance, for taking the time to review our manuscript, and for sharing her valuable insights and expertise.

## Author contributions

**Conceptualization:** Wanesa Wilczyńska.

**Data curation:** Wanesa Wilczyńska, Aymar Akilimali.

**Formal analysis:** Wanesa Wilczyńska.

**Investigation:** Wanesa Wilczyńska, Aymar Akilimali, Krzysztof Korzeniewski.

**Methodology:** Wanesa Wilczyńska.

**Project administration:** Krzysztof Korzeniewski.

**Software:** Wanesa Wilczyńska.

**Visualization:** Wanesa Wilczyńska.

**Writing – original draft:** Wanesa Wilczyńska.

**Writing – review & editing:** Aymar Akilimali, Krzysztof Korzeniewski.

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
