## [Decision Letter · Decision Letter 0]

21 Jan 2026

Human Filariasis in Africa (2000–2025): Changing Epidemiology, Uneven Diagnostic Progress, and Persistent Neglect

Dear Dr. Wilczyńska,

Thank you for submitting your manuscript to PLOS Neglected Tropical Diseases. After careful consideration, we feel that it has merit but does not fully meet PLOS Neglected Tropical Diseases's publication criteria as it currently stands. Therefore, we invite you to submit a revised version of the manuscript that addresses the points raised during the review process.

Please submit your revised manuscript within by Mar 22 2026 11:59PM. If you will need more time than this to complete your revisions, please reply to this message or contact the journal office at plosntds@plos.org. Please include the following items when submitting your revised manuscript:

We look forward to receiving your revised manuscript.

Kind regards,

Murtala Isah

Academic Editor

Guilherme Werneck

Section Editor

Shaden Kamhawi

co-Editor-in-Chief

Paul Brindley

co-Editor-in-Chief

**Journal Requirements:**

1) Please upload all main figures as separate Figure files in .tif or .eps format. For more information about how to convert and format your figure files please see our guidelines:

2) Some material included in your submission may be copyrighted. According to PLOSu2019s copyright policy, authors who use figures or other material (e.g., graphics, clipart, maps) from another author or copyright holder must demonstrate or obtain permission to publish this material under the Creative Commons Attribution 4.0 International (CC BY 4.0) License used by PLOS journals. Please closely review the details of PLOSu2019s copyright requirements here: PLOS Licenses and Copyright. If you need to request permissions from a copyright holder, you may use PLOS's Copyright Content Permission form.

Potential Copyright Issues:

- Figures 2, 3, 4, 5, and 6: Please (a) provide a direct link to the base layer of the map (i.e., the country or region border shape) and ensure this is also included in the figure legend; and (b) provide a link to the terms of use / license information for the base layer image or shapefile. We cannot publish proprietary or copyrighted maps (e.g. Google Maps, Mapquest) and the terms of use for your map base layer must be compatible with our CC BY 4.0 license.

**Reviewers' Comments:**

Reviewer's Responses to Questions

**Key Review Criteria Required for Acceptance?**

**Methods**

-Are the objectives of the study clearly articulated with a clear testable hypothesis stated?

-Is the study design appropriate to address the stated objectives?

-Is the population clearly described and appropriate for the hypothesis being tested?

-Is the sample size sufficient to ensure adequate power to address the hypothesis being tested?

-Were correct statistical analysis used to support conclusions?

-Are there concerns about ethical or regulatory requirements being met?

Reviewer #1: -Are the objectives of the study clearly articulated with a clear testable hypothesis stated?

Yes, the objectives of the study have been clearly articulated.

-Is the study design appropriate to address the stated objectives?

There is a big problem with the study design. Indeed, all entomological studies were excluded which was a big mistake because all these diseases are transmitted by vectors meaning that measuring the impact of interventions should not be based only on serology but should be confirmed by the results of the published entomology studies

-Is the population clearly described and appropriate for the hypothesis being tested?

-Is the sample size sufficient to ensure adequate power to address the hypothesis being tested?

I do not think that the sample size is high enough to come to the conclusion described in this study. So, as recommended in question above, I have strongly recommended to the authors to include several other studies based on entomology only or having both a serology and an entomology component. There are plenty published studies having both components not cited here.

-Were correct statistical analysis used to support conclusions?

Regarding this point, I have do not have the required skill.

-Are there concerns about ethical or regulatory requirements being met?

No, I do not think so.

Reviewer #2: Comments stated in reviewer's report

Reviewer #3: The aim of the the systematic review is well articulated, below are some suggestions for clarifying concepts and improving the article:

1.Clarify what study designs qualify (e.g., community household surveys, school-based surveys, sentinel villages, cluster-based elimination surveys). Specify whether cluster-based LF/OV programme surveys are included and why.

2. Explain how multiple datapoints from the same study/site/timepoint were handled when producing any overall summaries or trends, and describe rules for selecting or aggregating overlapping estimates.

3.The inclusion criteria require confirmatory diagnostics, yet exposure-based serology (e.g., Ov16 IgG4, Pan-IgG-SXP-1) is included. Define which tests were treated as evidence of active infection vs exposure, whether antigen tests were handled differently from antibody tests, and how heterogeneous endpoints were standardised. At minimum, categorise endpoints explicitly (mf microscopy, antigen detection, antibody/serology, molecular) and pre-specify how each contributes to analyses.

4.If metaprop was used, describe the meta-analytic model (e.g., random-effects), weighting, transformations, and heterogeneity outputs. If not, remove the pooled-estimate claim and describe a purely descriptive approach. Also specify whether 5-year means were sample-size-weighted or unweighted and justify the choice.

5.If linking prevalence declines to MDA, extract and incorporate programme exposure variables (years/rounds/coverage/baseline endemicity/program phase) or transparently classify settings as MDA-implemented vs no/unknown using study context and/or WHO/ESPEN data. Otherwise, soften causal language in Methods-driven interpretation.

Reviewer #4: (No Response)

**Results**

-Does the analysis presented match the analysis plan?

-Are the results clearly and completely presented?

-Are the figures (Tables, Images) of sufficient quality for clarity?

Reviewer #1: Regarding the results presented here, I can confirm that the analysis presented here matchs the analysis plan. The results are also really clear but for me these results are not presenting the real outcomes expected because plenty studies have been excluded.

Reviewer #2: Comments stated in reviewer's report

Reviewer #3: The results clearly presents the analysis plan, the limitations are clearly articluated. One limitation was dart of publications. Below are some suggestions for improvement:

6.Combining the mf prevalence, antigen prevalence, and antibody prevalence into the prevalence trend makes the outcome ambiguous. Present separate trend results by diagnostic endpoint (or stratified lines/sensitivity analyses) and interpret any combined line only as a heterogeneous average.

7.Figures currently show point estimates only; add confidence intervals/uncertainty bands (or IQRs/bootstrapped intervals if descriptive) and report the number of studies and total sample size contributing to each 5-year interval, especially important for sparse pathogens like L. loa and Mansonella.

8.If metaprop outputs were generated, present them (at least in supplementary forest plots by species × endpoint × interval) and report heterogeneity statistics; if not, remove references to pooled prevalence estimates.

9.Based on the extracted data, the results suggest there were “declines observed during the MDA era,” but they do not directly attribute these declines to MDA without specific programme exposure information. The wording should be adjusted to say “consistent with” or “occurring alongside.”

Reviewer #4: (No Response)

**Conclusions**

-Are the conclusions supported by the data presented?

-Are the limitations of analysis clearly described?

-Do the authors discuss how these data can be helpful to advance our understanding of the topic under study?

-Is public health relevance addressed?

Reviewer #1: The concludions presented in this paper are supported by the data presenetd here. However, we have several limitations. 1. a lot of published papers were not cited here. 2. entomology papers should be included as explained above and papers even containing both a serology and entomology components should be considered, which is not the case in the current evrsion of this manuscript.

Reviewer #2: Comments stated in reviewer's report

Reviewer #3: The conclusion supports data presented, except for the information on MDA which wa not part of the data extracted and analysed. The limitation is well articluated and the article is of public health relevance partiulary within the NTD community, highligtging the gap in survellience and diagnostics.

Reviewer #4: (No Response)

**Editorial and Data Presentation Modifications?**

Reviewer #1: Authors should include papers presenting only entomology data and also papers published with both serology and an entomology components.

Reviewer #2: Several critical issues require substantial revision before the manuscript can be considered for publication.

Reviewer #3: Typos/formatting: I noticed O. vulvulus (typo) in places (e.g., Fig 4 caption region). Please standardise to O. volvulus everywhere. Also, check for typos on the PRISMA flow.

Other typos; DISSCUSSION → DISCUSSION; Limitations of the review proces → process; several grammar issues (e.g., only 1 study were published; L. loa shows a increasing…).

Reviewer #4: (No Response)

**Summary and General Comments**

Reviewer #1: Though this is an important paper showing progresses made so far regarding the impact of interventions for the four filarial diseases, there are several limitations to address. Without addressing these limitations it will be challenging to support the conclusion made by the authors of this manuscript.

Other comment

Introduction

Burdens of onchocerciasis, Loa loa and Mansonellosis are missing. Please include them

Reviewer #2: Major comments

1. Lack of Quantitative Synthesis: No pooled estimates, confidence intervals, or measures of heterogeneity (I²) are presented in the main text or figures, despite the use of ‘metaprop’.

2. Alternatively, authors should provide forest plots or summary tables of pooled prevalence estimates for key species/periods in supplementary materials (not as a raw data in S2).

3. Formally test for subgroup differences (e.g., microscopy vs. molecular) using meta-regression, even if limited to more homogeneous subsets of studies. This is crucial to support claims that diagnostic method influences prevalence estimates.

4. The figures 3-6 show prevalence trends but without measures of variance (e.g., 95% CIs) or accounting for differing numbers of studies per period. At least, error bar should be added.

5. The pie charts for diagnostic methods should be accompanied by absolute numbers (n) in the legend or on the charts themselves.

6. The maps included in figure 3-6 required distinctive colour contrast and also needed to include the keys to interpret the colours.

7. The authors reported temporal trends, diagnostic methods, and geographic distribution of studies of species. However, geographical distribution should be a reflection of prevalence of the species in each region or country. This could be the ideal map for figures 3-6, instead of ‘geographical distribution of studies’. It is surprising that the authors did not analyze the prevalence of each species according to country or at least, representatives of top 5 or 10 countries with high prevalence should have been presented.

8. The discussion lacks depth and centers on mere description. The discussion describes the problems (geographic bias, diagnostic limitations) but falls short of providing a novel or actionable synthesis.

9. No indication in the manuscript that the study protocol was registered in PROSPERO. This is not acceptable.

Minor Comments

1. No clear objective(s) of the study stated in both abstract & introduction.

2. The keywords are redundant. Human filariasis; Neglected tropical diseases (NTDs); Africa; Lymphatic filariasis; Microscopy; Molecular diagnostics.

3. PECO Framework: The "Comparator" is vague. Specify that the comparison is implicit: trends over time and across diagnostic methodologies.

4. Clarify what constitutes a "data point." Is it a unique pathogen-study-population combination? Ensure this is consistently applied.

5. The authors need to avoid assertive statements on MDAs, since the study did not compare or report MDAs coverage or status for each region.

6. Limitations: ‘Many regions of Africa, particularly Northern and parts of Southern Africa, are underrepresented in indexed databases due to limited research infrastructure, scarce funding, and reduced opportunities for local investigators to publish in high-impact journals.’ This is a vague statement without any evidence to support.

Reviewer #3: This article is significant particularly for its focus on practical relevance and highlighting how limited surveillance and diagnostics can hinder elimination efforts and lead to incorrect interpretations of prevalence trends. It is notable for covering a broad 25-year period and analysing multiple pathogens, such as LF, onchocerciasis, loa-loa, and mansonellosis, especially considering their co-endemicity and potential diagnostic cross-reactivity.

Reviewer #4: This paper describes the changes in the epidemiology of filariases in Africa from 2020 to 2025. Overall the paper is well written and clearly described. Below are some comments to improve the content and relevance of the paper.

-When describing the human filariases, it is perhaps recommended to say loiasis and mansonellosis, in line with lymphatic filariasis and onchocerciasis. This should be addressed in the abstracts and throughout the text. If mentioning the scientific names of the parasites, then all should be mentioned. Not the disease in some cases and the parasites in others.

- Lines 69 - 72. It might be useful to indicate the different Mansonella species known to cause human disease in Africa.

- Line 99: L. loa

- Figure 1: Check the spelling of "papers" and "sufficient".

- Line 175: Please provide the exact number of participants. Also, it is not clear what the data points refer to. Please clarify.

- Figure 2: A colored image would be helpful. Kindly reverse the colors for the countries with little data to have the lighter colors and those with more data to have the deeper colors.

- Figures 3 - 6: A legend should be provided for the maps. Same comments on the colors also apply here.

- Result on Mansonella spp.: Could these be further elaborated to indicate the species that are most common?

- Line 254: Check the spelling of "Discussion"

- Results: it would have been useful to present a country by country analysis of each disease and the decline in prevalence. This would be important for disease control programmes.

- It would be very interesting and useful to know how the results compare to the ESPEN data.

-

PLOS authors have the option to publish the peer review history of their article (what does this mean?). If published, this will include your full peer review and any attached files.). If published, this will include your full peer review and any attached files.). If published, this will include your full peer review and any attached files.). If published, this will include your full peer review and any attached files.

...

Reviewer #1: No

Reviewer #2: **Yes:** OLAWALE QUAZIM JUNAIDOLAWALE QUAZIM JUNAIDOLAWALE QUAZIM JUNAIDOLAWALE QUAZIM JUNAID

Reviewer #3: No

Reviewer #4: No

**Figure resubmission:**While revising your submission, we strongly recommend that you use PLOS’s NAAS tool (https://ngplosjournals.pagemajik.ai/artanalysis) to test your figure files. NAAS can convert your figure files to the TIFF file type and meet basic requirements (such as print size, resolution), or provide you with a report on issues that do not meet our requirements and that NAAS cannot fix.
---

## [Decision Letter · Decision Letter 1]

25 Mar 2026

Dear phD Wilczyńska,

We are pleased to inform you that your manuscript 'Human Filariasis in Africa (2000–2025): Changing Epidemiology, Uneven Diagnostic Progress, and Persistent Neglect' has been provisionally accepted for publication in PLOS Neglected Tropical Diseases.

Best regards,

Sarman Singh, MD, FRSC, FRCP

Section Editor

Claudia Brodskyn

Section Editor

Shaden Kamhawi

co-Editor-in-Chief

Paul Brindley

co-Editor-in-Chief

Reviewer's Responses to Questions

**Key Review Criteria Required for Acceptance?**

**Methods**

-Are the objectives of the study clearly articulated with a clear testable hypothesis stated?

-Is the study design appropriate to address the stated objectives?

-Is the population clearly described and appropriate for the hypothesis being tested?

-Is the sample size sufficient to ensure adequate power to address the hypothesis being tested?

-Were correct statistical analysis used to support conclusions?

-Are there concerns about ethical or regulatory requirements being met?

Reviewer #2: (No Response)

**Results**

-Does the analysis presented match the analysis plan?

-Are the results clearly and completely presented?

-Are the figures (Tables, Images) of sufficient quality for clarity?

Reviewer #2: (No Response)

**Conclusions**

-Are the conclusions supported by the data presented?

-Are the limitations of analysis clearly described?

-Do the authors discuss how these data can be helpful to advance our understanding of the topic under study?

-Is public health relevance addressed?

Reviewer #2: (No Response)

**Editorial and Data Presentation Modifications?**

Reviewer #2: (No Response)

**Summary and General Comments**

Reviewer #2: (No Response)

PLOS authors have the option to publish the peer review history of their article (what does this mean?). If published, this will include your full peer review and any attached files.). If published, this will include your full peer review and any attached files.). If published, this will include your full peer review and any attached files.). If published, this will include your full peer review and any attached files.

...

Reviewer #2: **Yes:** OLAWALE QUAZIM JUNAIDOLAWALE QUAZIM JUNAIDOLAWALE QUAZIM JUNAIDOLAWALE QUAZIM JUNAID

---

## [Editor Report · Acceptance letter]

Dear phD Wilczyńska,

We are delighted to inform you that your manuscript, "Human Filariasis in Africa (2000–2025): Changing Epidemiology, Uneven Diagnostic Progress, and Persistent Neglect," has been formally accepted for publication in PLOS Neglected Tropical Diseases.

Best regards,

Shaden Kamhawi

co-Editor-in-Chief

Paul Brindley

co-Editor-in-Chief
